# Do medical specialists accept claims-based Audit and Feedback for quality improvement? A focus group study

Vera de Weerdt [1,2] Sierk Ybema,[3,4] Sjoerd Repping,[2] Eric van der Hijden,[1,5] Hanna Willems[6]

¹Department of Health Economics, Vrije Universiteit Amsterdam, Amsterdam, The Netherlands
²Amsterdam University Medical Centres, Amsterdam, The Netherlands
³Department of Organization Sciences, Vrije Universiteit Amsterdam, Amsterdam, The Netherlands
⁴Department of Organization Sciences, Anglia Ruskin University, Chelmsford, UK
⁵Zilveren Kruis Health Insurance, Leiden, The Netherlands
⁶Department of Geriatrics, Amsterdam University Medical Centres, Amsterdam, The Netherlands

**Correspondence to**
Vera de Weerdt;
v.de.weerdt@vu.nl

## ABSTRACT

**Objectives** Audit and Feedback (A&F) is a widely used quality improvement (QI) intervention in healthcare. However, not all feedback is accepted by professionals. While claims-based feedback has been previously used for A&F interventions, its acceptance by medical specialists is largely unknown. This study examined medical specialists' acceptance of claims-based A&F for QI.

**Design** Qualitative design, with focus group discussions. Transcripts were analysed using discourse analysis.

**Setting and participants** A total of five online focus group discussions were conducted between April 2021 and September 2022 with 21 medical specialists from varying specialties (urology; paediatric surgery; gynaecology; vascular surgery; orthopaedics and trauma surgery) working in academic or regional hospitals in the Netherlands.

**Results** Participants described mixed views on using claims-based A&F for QI. Arguments mentioned in favour were (1) A&F stimulates reflective learning and improvement and (2) claims-based A&F is more reliable than other A&F. Arguments in opposition were that (1) A&F is insufficient to create behavioural change; (2) A&F lacks clinically meaningful interpretation; (3) claims data are invalid for feedback on QI; (4) claims-based A&F is unreliable and (5) A&F may be misused by health insurers. Furthermore, participants described several conditions for the implementation of A&F which shape their acceptance.

**Conclusions** Using claims-based A&F for QI is, for some clinical topics and under certain conditions, accepted by medical specialists. Acceptance of claims-based A&F can be shaped by how A&F is implemented into clinical practice. When designing A&F for QI, it should be considered whether claims data, as the most resource-efficient data source, can be used or whether it is necessary to collect more specific data.

## INTRODUCTION

Audit and Feedback (A&F) interventions are widely used as a quality improvement (QI) strategy in healthcare.[1] Through A&F, medical professionals receive feedback on their clinical performance, which intends to stimulate them to improve their quality of care.[2–4] Using readily available claims data for A&F aiming to improve quality has large potential benefits, since it can prevent administrative burdens on professionals and limit costs associated with data collection for A&F.[5 6]

However, a review on A&F from registries suggests that claims-based A&F may not be suitable for QI interventions, because the use of claims-based A&F was associated with a lower effect on QI as opposed to A&F from quality registries.[7] Healthcare claims data are collected for the purpose of reimbursement of healthcare services. The data contain patients' personal information, a diagnosis code and billing codes for provided services and medications.[8] As claims data are not collected for clinical decision making or QI, it often lacks clinical detail, information on disease severity and relevant patient characteristics such as smoking status.[5 9 10] Therefore, clinicians may not accept claims-based A&F as 'fit-for-purpose' in the context of QI interventions.[11 12]

Research on whether medical specialists accept the use of claims-based A&F in the context of QI is limited. The aim of this study was to examine whether medical specialists accept claims-based A&F intended for QI purposes.

## METHOD
### Study design

We conducted focus groups with medical specialists from varying specialties, to examine

their potential acceptance of claims-based A&F. Focus groups were used to allow for discussions between participants on varying perceptions and arguments, thereby creating richness of results.[13]

### STRENGTHS AND LIMITATIONS OF THIS STUDY

⇒ Focus on the perspective of medical specialists on claims-based Audit and Feedback (A&F).
⇒ Comprehensive findings due to the inclusion of medical specialists from varying specialties.
⇒ Discourse analysis to create an in-depth understanding of participants' views on claims-based A&F.
⇒ Reflection on examples of Dutch claims-based A&F and Dutch claims data.
⇒ No inclusion of non-surgical specialties.

This study was designed in accordance with the COnsolidated criteria for REporting Qualitative research (online supplemental file 1).[14] All participants provided written informed consent.

### Patient or public involvement
Patients or the public were not involved in this study.

### Contextual background of this study
In a previous study, claims-based A&F was developed in a co-creation approach with six Comparative Effectiveness Research (CER) teams (online supplemental file 2). These CER studies are conducted to identify which interventions are most effective or most cost-effective and the aim is to subsequently implement the CER results in practice to create QI. A&F in this context portrays the level in which CER results are implemented into clinical practice, which is intended to stimulate professionals to implement CER results where necessary.

In the current study, we used this previously developed claims-based A&F to explore whether independent medical specialists, that is who were not involved in the A&F development process, accept claims-based A&F for QI purposes.

### Participants
For each focus group, we invited medical specialists working in the specialty of the respective CER study, namely: urology, paediatric surgery, gynaecology, orthopaedics & trauma surgery, and vascular surgery. Recruitment of medical specialists was done between March 2021 and June 2022 in two ways: first, we directly invited medical specialists of respective specialisms via email. Email addresses were obtained via the CER study teams or via the professional network of the authors. Second, some of the CER study teams also contacted their professional networks via email and/or newsletters. The reasons for doing this study were explained in the recruitment email and at the start of each focus group.

In online focus groups, it is recommended to use a smaller number of participants to ensure active participation, thus we aimed for 4–6 participants per session.[15] We aimed to recruit a diverse group of participants in characteristics like gender, years of work experience, geographical region and type of hospital. We invited 93 medical specialists directly, of which 29 did not respond to the invitation and 10 refused to participate for one of the following reasons: lack of time (n=3), did not consider themselves an expert on the topic (n=5) or no reason (n=2). Of the 54 medical specialists who were interested in participation, 33 were unable to participate on the scheduled dates of the focus groups.

In one of the focus groups (DART), three of the eight participants were previously familiar with each other

through professional connections, in the other four focus groups none of the participants had previous relationships.

### Data collection
Focus groups were held online via Microsoft Teams between April 2021 and September 2022. Focus groups were held during working hours and lasted 60–90 min. Researcher HW (MD, PhD, female) or SY (PhD in organisation sciences, male) moderated the sessions. Researcher VdW (MD, PhD candidate, female) acted as an observer, presented the examples of claims-based A&F for CER trials and asked clarifying questions when necessary. Both moderators had extensive experience with focus group research and moderation of focus groups. The observer had prior experience with qualitative research, but not with moderation of focus groups. Three participants of one focus group (DART) had a previous professional connection to one of the moderators, who had a limited supporting moderation role in this focus group. None of the other participants had previous connections to the researchers. The moderators of the focus groups were not involved in the co-creation study in which the examples of claims-based A&F for CER studies were developed.

Each focus group followed the same protocol (online supplemental file 3). Focus groups consisted of two parts. First, we asked participants whether they were familiar with A&F for QI and whether they were familiar with claims-based A&F - to elicit participants' general perceptions on A&F and claims-based A&F. Second, we presented participants with the specific examples of claims-based A&F (online supplemental file 4). These examples were visual graphs of claims-based A&F for one specific CER study, which presented the volume of the intervention studied in the CER study, per hospital in the year 2017 (figure 1). This enabled us to capture participants' perceptions on realistic examples of claims-based A&F.

Focus groups were audio-recorded or video-recorded and subsequently transcribed ad verbatim. Field notes were taken by the observer during each focus group. Transcripts were not returned to participants. No repeat focus groups were carried out.

### Data analysis
Transcripts were analysed using discourse analysis.[16] Discourse analysis can be useful for creating an in-depth understanding of how people construct contested issues in different ways.[16 17] Discourse analysis can then reveal rhetorical devices people use to make 'what they are saying appear factual'.

Authors VdW and HW independently coded each transcript after a focus group was held, coding was done in Microsoft Word or MAXQDA. To ensure coding reliability, they discussed the coding of each transcript and resolved discrepancies through discussion and refined codes. In the fifth and final focus group, no new codes were identified, which provided evidence for data saturation. After coding the final focus group, codes were discussed within

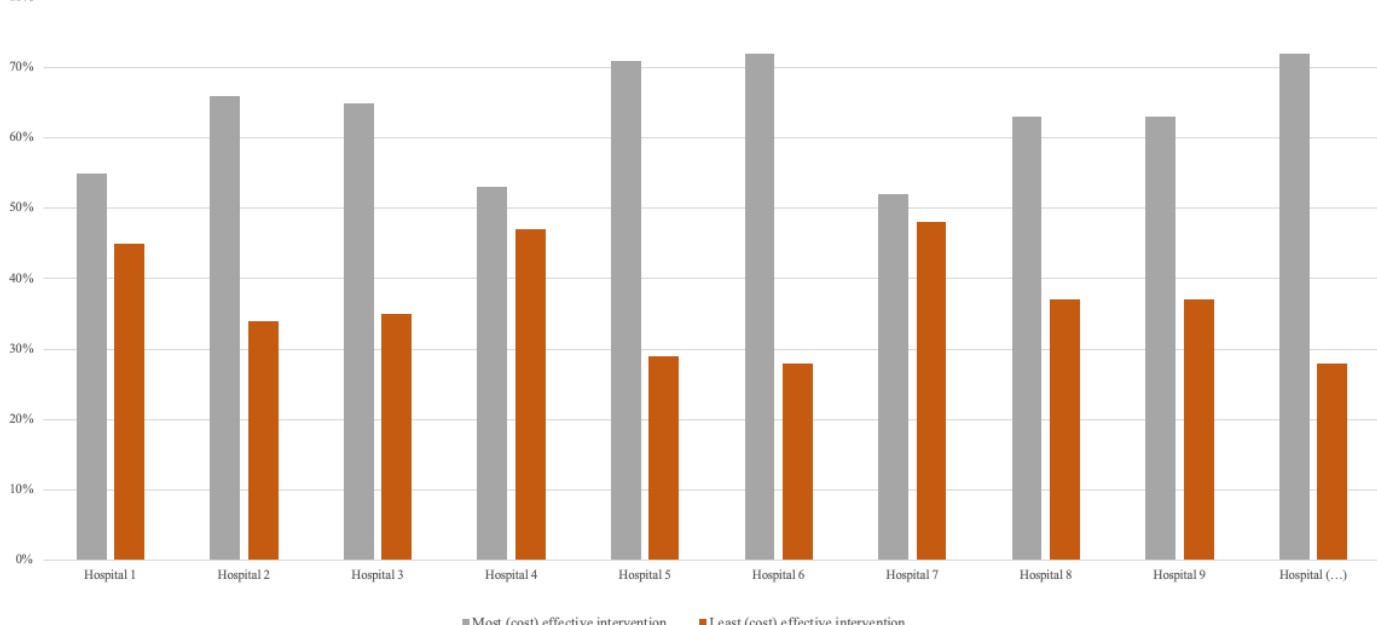

**Figure 1** Hypothetical example of claims-based A&F on CER implementation level per hospital. A&F: Audit and Feedback; CER: Comparative Effectiveness Research.

the wider research team (VdW, HW and SY). The wider research team decided to group codes into arguments in favour and against claims-based A&F. Then, authors VdW and HW re-analysed each transcript and labelled all codes as arguments in favour or against claims-based A&F and identified supporting quotes. Again, authors VdW and HW discussed labels of each argument and resolved discrepancies through discussion. Subsequently, author VdW grouped arguments and supporting quotes. Last, quotes were analysed using a discourse analytic approach to formulate how participants discursively presented their arguments. This was done by analysing key quotes and singling out the figures of speech and key words participants used in order to make a case for or against claims-based A&F (e.g., 'it works, because it presents a mirror'). On the basis of this first-order analysis, the underlying discursive strategies which participants adopted (e.g., presenting claims-based A&F as an effective tool for QI) were analysed. These strategies were discussed within the wider research team (VdW, HW, SY) until all authors agreed on the formulation of the arguments.

## RESULTS
### Participants
In total, there were 21 participants from six clinical specialties from 17 hospitals in the Netherlands (table 1). The number of participants per focus group varied from 2 to 7. Most participants described familiarity with A&F, with 62% describing regular use, and an additional 19% indicating some level of exposure. Conversely, 14% reported no prior experience with A&F. Specifically focusing on

claims-based A&F, 38% of participants described regular use, 14% described some level of exposure and 48% described no prior experience with this type of A&F.

### Acceptance of claims-based A&F
Most participants were positive in their views on A&F in general, but participants were mixed in their views on claims-based A&F and the specific examples of claims-based A&F for CER studies. While some participants were unambiguously in favour or against claims-based A&F, most participants posed arguments for both sides (online supplemental files 5 and 6). Participants who were familiar with A&F were more positive towards claims-based A&F than participants who were unfamiliar with A&F. Furthermore, participants who were in favour of A&F for QI argued their stance depended on how claims-based A&F was implemented in practice.

#### Arguments in favour of claims-based A&F
*A&F stimulates reflective learning and improvement*
First, participants who viewed A&F favourably described it as a tool that creates reflective learning and improvement. They believed A&F could contribute to QI by providing feedback on whether CER results are implemented into clinical practice. Participants described the presented visual graphs of claims-based A&F as a 'visually attractive tool' to initiate change. They believed feedback could stimulate a conversation on change. To explain their positive assessment, they used metaphors to frame A&F as a tool that activates and accelerates change: 'A&F is setting everything in motion', and as an instrument that stimulates self-critical reflection: 'Showing people a mirror is

**Table 1** Participant characteristics

| Focus group | Participant | Function | Type of hospital | Gender | Familiar with A&F | Familiar with A&F based on claims data |
|---|---|---|---|---|---|---|
| DART | D1 | Trauma surgeon | Top-clinical | M | Somewhat | No |
| | D2 | Trauma surgeon | Top-clinical | M | Somewhat | Somewhat |
| | D3 | Orthopaedic surgeon, traumatologist | Top-clinical | M | Somewhat | No |
| | D4 | Trauma surgeon | Top-clinical | M | No | No |
| | D5 | Trauma surgeon | Academic | M | Yes | No |
| | D6 | Orthopaedic trauma surgeon | Top-clinical | M | No | Somewhat |
| | D7 | Orthopaedic trauma surgeon | Academic | M | No | No |
| STONE | S1 | Urologist | Top-clinical | F | Yes | Yes |
| | S2 | Urologist | Top-clinical | M | Yes | Yes |
| | S3 | Urologist | Academic | M | Yes | Yes |
| | S4 | Urologist | General | F | No | No |
| Proclion | P1 | Vascular surgeon | Top-clinical | M | Somewhat | Somewhat |
| | P2 | Vascular surgeon | Top-clinical | M | Yes | No |
| MIRA2 | M1 | Gynaecologist | General | F | Yes | Yes |
| | M2 | Gynaecologist | Top-clinical | F | Yes | Yes |
| | M3 | Gynaecologist | Academic | M | Yes | Yes |
| | M4 | Gynaecologist | Academic | F | Yes | Yes |
| | M5 | Gynaecologist | General | F | Yes | Yes |
| CAPP | C1 | Paediatric surgeon | Academic | M | Yes | No |
| | C2 | Paediatric surgeon | Top-clinical | F | Yes | No |
| | C3 | Paediatric surgeon | Top-clinical | M | Yes | No |

*In the Netherlands, general hospitals are smaller, regional hospitals which are most often non-teaching hospitals. Top-clinical hospitals distinguish themselves from general hospitals by meeting certain criteria, which include teaching programs, an infrastructure for innovation and scientific research, and excelling in areas of clinical care. Academic hospitals are large teaching and research hospitals, which provide tertiary care.[32]

why it works', because, as another participant put it: 'By comparing yourself you gain insights into whether you are doing 'the right thing". Some participants described that they believe A&F is mostly useful for outliers. The imagery some participants used reflected their positive reception of A&F as a tool for change.

*Claims-based A&F is more reliable than other types of A&F*
A second argument in favour of claims-based A&F was that it is more reliable than other types of A&F, as claims data are perceived as more reliable than self-reported data. Participants described:

> If you can get it out of claims data, then you have objective data on your own process, because otherwise it is dependent on what the person registers and that can be less reliable.

> In quality-registry data it depends on the administrator whether A&F is accurate, if you can improve that by claims-based A&F, then you have a better and more accurate registration, which gives better A&F.

Here, participants described self-reported data negatively by labelling it as potentially 'less reliable' or 'inaccurate', while they described claims data as 'objective' and 'more accurate'. These participants posed that 'objective' and 'more accurate' claims data leads to 'better A&F'.

*Arguments against claims-based A&F*
*A&F is insufficient to create change of clinical behaviour*
A first counterargument to use A&F was that it is insufficient to create change of clinical behaviour. Some participants questioned the effectiveness of A&F:

> I wonder then if you would have the same data and would show it to hospitals if that would really lead to change of practice. I don't know yet. Maybe they will still think of explanations for why they deviate from the norm.

Another participant describes the insufficiency of A&F as:

> If you get the A&F, you need to act upon it, but you can't reinforce that, so… there is the dilemma: you

de Weerdt V, *et al. BMJ Open* 2024;**14**:e081063. doi:10.1136/bmjopen-2023-081063

can show it, but subsequently you want change, but that change depends on individual hospitals, so….

Keywords are 'dilemma' and 'reinforce'. The speaker hereby assumes that for behavioural change by professionals to happen this must be enforced, which A&F itself does not. Another participant described how A&F does not create the intended QI for him, but it only incentivises him to register claims differently:

If you have the feeling you are doing evidence-based treatments and that all it takes is to register claims codes differently, then you will change claims codes: we are not more catholic than the pope.

Here, the participant used the metaphor 'we are not more catholic than the pope' to describe that medical specialists are not going to strictly abide by the rules to achieve the goal set by A&F, but rather find a loophole to meet the goal without having to change their clinical behaviour.

Thus, participants described the insufficiency of A&F to create QI, by questioning its effectiveness or by contriving explanations for why A&F will not lead to QI.

### Claims-based A&F lacks clinically meaningful interpretation

A second argument against A&F was that participants felt it was not possible to interpret the visual graphs of claims-based A&F in a clinically meaningful way. Participants questioned the benefits of the data: 'Yes, it is a nice overview. Only, ehm yes, what are the consequences of this figures? I can't … ehm… I can't see how to translate this to my practice? […] How do we or the patient benefit from this exactly?'. Furthermore, some participants stated more strongly that if they could not interpret the data they would not use it: 'I can't explain these results, thus then I would discard the feedback'.

### Claims data give an invalid representation of clinical reality

The third counterargument was that claims data give an invalid representation of clinical reality. When presented with the visual graphs of claims-based A&F on CER, for three out of five cases, the participants described the visual graphs negatively: 'It is comparing apples and oranges' or 'You will receive critic of it is not specific enough, thus I can't do anything with it (…) it is caused by that stupid claims system, it is just not designed properly for this'. Using the metaphor 'comparing apples and oranges' shows the participant believed an unjust comparison is made, in this case the claims-based A&F which is not a valid comparison with the CER population. Furthermore, 'not designed … for this' also describes that the participant believed claims data are used for a purpose (A&F on CER), for which it is not suitable or valid. Use of the word 'stupid' also shows the participants' frustration with the claims system. Thus, for three out of five cases, the participants described the claims-based A&F as invalid feedback on CER.

### Claims data give unreliable results on the number of performed interventions

A fourth argument against claims-based A&F was that claims data give unreliable results on the number of performed interventions, thus claims-based A&F is not reliable either. Participants described personal experiences with registration of claims data and with claims-based A&F to illustrate their point:

Claims diagnoses may be inaccurate since they are registered prior to diagnostic interventions and they are often not adjusted after an alternative diagnosis is identified through diagnostic testing.

In our own registration we have conducted a procedure a 100 times and claims data only shows 90 procedures, then I think where did the other 10 go?

Participants emphasised their mistrust of claims data by stating:

Over the years I have started to greatly mistrust claims data.

If you want to pull data from the claims system, the coding can go wrong and then you get a group out of your search which is not representative of what you actually did (…) you need to be very careful with this: you may think you see everything, but you're only using a flashlight in a dark room.

Here participants used the word 'greatly' to emphasise their mistrust in claims data and the metaphor 'only using a flashlight in a dark room' to illustrate their perception that claims-based A&F can only show a small part of clinical practice, which may lead to incorrect judgements about clinical performance.

Thus, the participants used anecdotal experiences to convey objectiveness in their arguments against claims-based A&F and figures of speech to emphasise their objections of claims-based A&F.

### Claims-based A&F may be misused by health insurers

The last argument against claims-based A&F was that participants feared the role of the health insurer. Participants described that they feel health insurers may have a conflict of interest, such as financial incentives to decrease volumes of care, instead of QI purposes. 'If it is the health insurer who has all that claims data, then they have an instrument to say - maybe on wrong grounds - you operate too much or you operate too little' and 'or even worse, that the health insurer will say to me, you can't operate anymore, because that is too expensive'.

Keywords here are 'wrong grounds', in which the speaker assumed that health insurers make decisions based on wrong grounds, and 'even worse' to emphasise the severity of their argument.

Furthermore, participants described that they fear the role of health insurers when using claims-based A&F:

What I fear…. is that it won't be used for the patient, but that it will be used by health insurers to condemn hospitals and that …it is tricky this.

Yes, I greatly fear the judgement of the health insurer who is just looking over my shoulder, even though they might be right in the end that we overtreated, but yeah.

There are always fearful images - if we operate 60% and my neighbour only operates 20% of patients - that the health insurer will come to me saying 'you operate a bit much, that is a little bit expensive, are you performing procedures correctly?' So yeah, claims data is always a bit of a tricky thing (…) I think claims data may be a thing we should avoid.

Here, the participants legitimised their objections to claims-based A&F by posing that the role of health insurers is something that should be 'feared'.

Furthermore, participants described that they do not want to justify their behaviour to the health insurer, as this leads to less joy in work:

And if a health insurer says, (…) you have to do it this way or why did you do that, then I have to justify myself to the health insurer, which will make the work less fun.

Thus, the participants used several types of discourse to describe the role of health insurers in claims-based A&F as negative.

### Implementation of claims-based A&F in practice
*A&F only suitable for topics in which sufficient scientific evidence is available*
Participants argued that A&F is only suitable for topics on which scientific evidence is available for what is considered 'best practice'. To emphasise their argument, they described using A&F without scientific evidence as 'dangerous':

For wrist fractures we have long not figured out what is best for the patient, we do have ideas, but we have long not figured it out, so then the use of A&F could be dangerous.

Another participant described:

Well, yeah, you should not give A&F on topics of which the truth is unknown yet, so you should not confuse the opinion of the majority with the truth.

This participant described scientific evidence as 'the truth' and posed that A&F on topics without scientific evidence may cause bias in misconceiving the clinical practice of the majority is 'the best'.

*Distribution through professional association creates support for A&F among medical specialists*
Furthermore, participants described that distribution of A&F done via their professional associations, instead of via health insurers, is more likely to be accepted:

I do agree that if A&F comes through our professional association or guideline committee and it is included in our own guideline that there will just be much more support for it.

A participant emphasised this point by stating:

Because yes, that (the clinical guideline) is just our bible.

The participant used the word 'bible' to emphasise the value of and support for clinical guidelines of professional associations.

However, other participants stated that who distributes the data is unimportant to them:

As long as we can assume the A&F is valid, then I don't care if the insurer or the hospital distributes it.

*Function of A&F should not be judgement of performance*
Lastly, participants argued claims-based A&F should not be used to judge whether clinical performance is 'wrong' or 'right':

but to fully say, here we put the division line, these hospitals are doing right and these are doing wrong based on this type of data, we should stay far away from that, I feel.

We should use A&F, but we should not 'name and shame', that is what I want to stay away from.

Here participants' use of the words 'division line' and 'name and shame' showed their negative perceptions on judgement of performance based on A&F.

### DISCUSSION
This study examined medical specialists' acceptance of claims-based A&F for QI through focus group discussions, in which participants were presented with examples of claims-based A&F on CER trials. While participants were mostly positive in their views on A&F in general, they were mixed in their views on claims-based A&F and on the specific examples of claims-based A&F for CER trials. Most participants put forward arguments both in favour and against claims-based A&F. Arguments in favour of claims-based A&F were that (1) A&F stimulates reflective learning and improvement and (2) claims-based A&F is more reliable than other A&F. The opposition against A&F as a QI tool was articulated in five ways: participants claimed that (1) A&F is insufficient to create change of clinical behaviour; (2) A&F lacks clinically meaningful interpretation; (3) claims data are perceived as invalid for feedback on QI; (4) claims-based A&F is perceived as unreliable and (5) claims-based A&F may be misused by health insurers. Additionally, participants indicated that their acceptance depended on how claims-based A&F would be implemented: (1) there should be sufficient clinical evidence for the topic of A&F; (2) A&F

should be distributed by professional societies and (3) A&F should not be used to impose judgement on clinical performance.

Participants had varying views on the effectiveness of A&F as a tool to create QI. We observed that participants without prior use of A&F were more sceptical of A&F as a tool than participants who had used A&F prior to the focus group. In line with previous research, this suggests being unknown with A&F, makes it unloved.[18 19] While A&F has been used as a QI intervention in healthcare for years, studies consistently report physicians' incapability to interpret population-level feedback.[19–22] This leads to negative perceptions regarding the usefulness of A&F, which decreases acceptance of A&F.[19] Thus, our findings seem to corroborate earlier findings that physicians should be trained on how to interpret and use A&F for QI.[19 21] However, our findings also suggest that acceptance of A&F does not depend solely on informing and training physicians.

Data validity and reliability was mentioned by participants as the most important factor to accept claims-based A&F for QI purposes. Again, the participants gave mixed reactions, some viewed claims data as potentially more reliable than self-reported data, while others questioned the reliability of claims data and the validity of claims-based A&F for QI. Studies examining acceptance of A&F consistently report data quality as the main factor for accepting A&F, irrespective of which data source is used for the feedback.[3 21–25] Other studies on claims-based A&F found physicians had mainly negative perceptions on the quality of claims data,[11 21 22 26] while we also find positive views. For claims-based A&F, it is important to note that the validity and reliability are dependent on the clinical topic for which A&F is used. For example, claims data capture whether a patient had a wrist fracture, but lack detail of the location of the fracture. This detail may not be relevant for reimbursement but can be clinically relevant in the context of QI, thereby decreasing the 'fit-for-purpose' of using claims data for A&F on QI. The validity or 'fit-for-purpose' for A&F on QI is determined by how specific claims data distinguishes clinically relevant patient groups. For some healthcare fields, claims data are more detailed than others, increasing the 'fit-for-purpose' of claims-based A&F for QI within these specialties.[27] The reliability of claims data may also vary for clinical topics: participants in our study noted that claims data are registered more reliable for larger procedures (e.g., surgeries) as compared with smaller interventions and diagnoses for acute, conservatively treated disorders. Thus, the applicability of claims data for A&F in QI is contingent on the clinical topic.

While lack of data quality is consistently mentioned as a barrier for using A&F, it is important to note that these objections are not only about the data itself, but professionals' perceptions play a big role in accepting the data as accurate or not. For example, data accuracy is questioned more strongly when A&F shows performance is low or when performance is judged.[3 24] Data accuracy

may also be less questioned in A&F interventions that are voluntary and/or physician-led.[19 28]

Participants in our study suggested to distribute claims-based A&F via professional associations, instead of via health insurers. This is in line with other research, which finds A&F is more accepted and effective if it comes from a colleague instead of an external.[1 3] While endorsement of A&F by professional associations could be beneficial in its acceptance by physicians, studies suggest that A&F is most effective if it is given by a direct colleague.[1 29] Thus, it would be interesting to examine whether endorsement of A&F by professional associations is sufficient to create QI or whether local champions of A&F are necessary to create change.

In our study, we only examined medical specialists' acceptance of the data source used for A&F: claims data. However, acceptance of claims data as the source for A&F does not guarantee medical specialists' acceptance of an A&F intervention. This is also dependent on the content of the feedback (e.g., is it timely, personalised); recipient characteristics (e.g., do they have an internal/external locus of control) and the audit method (e.g., was it transparent).[3 21 23 30] One study posed acceptance of A&F is unnecessary and that the goal of changing clinical behaviour can also be achieved by combining A&F with for example financial incentives.[31] However, lack of physicians' acceptance of A&F will likely not create a sustainable QI culture among physicians.

Our study suggests that, under certain conditions, medical specialists can accept claims-based A&F for QI purposes. In the review of van de Veer *et al*, it is uncertain whether the source of A&F being claims data are the explaining factor for lower effectiveness of the interventions.[7] Even if claims data are not the most effective data source, it is currently the most resource-efficient data source for A&F interventions, especially for large-scale interventions aiming to provide A&F to multiple providers. Thus, policymakers, managers and professionals should weigh whether the clinical topic can be portrayed by claims-based A&F or whether it is necessary to collect more detailed data for A&F aiming to improve quality.

### Strengths and weaknesses

This study examined medical specialists' acceptance of claims-based A&F in the context of QI. The first strength of this study is the discourse analysis through which we were able to elicit medical specialists' mixed views on claims-based A&F, which creates a better understanding of medical specialists' acceptance of claims-based A&F and shows that it is not simply the reliability of the data that determines the acceptability of claims-based A&F, but rather physicians' perceptions of its reliability. Second, to the best of our knowledge, this is the first study on claims-based A&F focussing on the perceptions of medical specialists. Another strength of this study is that we used realistic examples of claims-based A&F on varying CER studies and included medical specialists from various clinical specialities. Furthermore, we included both participants who were and were not familiar

with A&F and claims-based A&F, which increases the generalisability of our results.

A limitation of this study is that participants in our study reflected on Dutch claims data. As the content of claims data and the accuracy of registration can vary by country, medical specialists in other countries may view claims-based A&F differently. However, our results describe themes which are common across healthcare systems and our results corroborate and detail previous findings described in A&F literature. Second, the participants in our focus groups were mainly surgeons, thus it is unclear whether other healthcare professionals put forward additional perspectives on claims-based A&F. Lastly, in some focus groups we had a limited number of participants which may have decreased the level of discussion between participants. However, we did reach data saturation over all focus groups, thus we do not believe that additional arguments would have been identified with more participants per focus group.

## CONCLUSION

Using claims-based A&F for QI is, for some clinical topics and under certain conditions, accepted by medical specialists. Our study showed that perceptions of medical specialists on A&F in general and claims-based A&F for QI varied, and acceptance of claims-based A&F can be shaped by how A&F is implemented into clinical practice. Currently, claims data are the most resource-efficient data source for A&F interventions. Thus, policymakers, managers and professionals should consider whether the clinical topic can be represented by claims-based A&F or whether it is necessary to collect other data for A&F with the aim of improving quality.

**Contributors** HW, VdW and EvdH designed the study. SY was involved in the study for his extensive expertise in qualitative research and discourse analysis. HW and/or SY moderated the focus groups. VdW was present in the focus groups as an observer and presenter. SR provided overall supervision and was involved in the write-up of this study. All authors were involved in the write-up of this study. HW is the guarantor of this study.

**Funding** This research was carried out within the Academic Working Place Care Practice and Policy of the Consortium Quality of Care of the Netherlands Federation of University Medical Centers (NFU), and National Health Care Institute. This funder had no role in the design of the study, data collection, data analysis, interpretation of data and writing the manuscript.

**Competing interests** None declared.

**Patient and public involvement** Patients and/or the public were not involved in the design, or conduct, or reporting, or dissemination plans of this research.

**Patient consent for publication** Not applicable.

**Ethics approval** This study involves human participants. This research was conducted in accordance with the ethical guidelines of the Research Ethics Review Committee (BETHCIE) of the Faculty of Science, Vrije Universiteit. All participants received an information letter stating the reasons for conducting the research and provided written informed consent for participation in this study and publication of results. Since the research did not fall under the Medical Research Involving Human Subjects Act (WMO), the study protocol was self-checked by the Research Ethics review committee of the Faculty of Science (BETHCIE) from the VU Amsterdam Faculty of Science, stating that the research project did not require further evaluation by the Research Ethics Review Committee of the VU Amsterdam Faculty of Science, exempted this study. Participants gave informed consent to participate in the study before taking part.

**Provenance and peer review** Not commissioned; externally peer reviewed.

**Data availability statement** Data are available upon reasonable request. The data generated and analysed during the current study are not publicly available since these include recordings of focus group sessions with individual persons, but transcripts are available from the corresponding author upon reasonable request.

**ORCID iD**
Vera de Weerdt http://orcid.org/0000-0003-3644-620X

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
