## [Reviewer comments · BMJ Open]

ARTICLE DETAILS

TITLE (PROVISIONAL)	Do medical specialists accept claims-based Audit & Feedback for quality improvement? a focus group study
AUTHORS	de Weerd, Vera; Ybema, Sierk; Repping, S.; van der Hijden, Eric; Willems, Hanna

VERSION 1 – REVIEW

REVIEWER	González, Ricardo Espinoza Clínica de la Universidad de los Andes
REVIEW RETURNED	13-Nov-2023

GENERAL COMMENTS	This study represent a contribution in the quality improvement strategy. They ask specialist in different fields of surgery, where the claims-based analysis is very important. The possible limitation for use only Dutch data I believe is secondary because the reality is similar in other countries.
---

REVIEWER	Sharma, Megha Karolinska Institutet, Department of Global Public Health
REVIEW RETURNED	21-Nov-2023

GENERAL COMMENTS	Do medical specialists accept claims-based Audit & Feedback for quality improvement? a focus group study Thank you for presenting this manuscript focusing on the issue of acceptance of claim-based audit and feedback among medical specialists. It is an important topic, which is usually not focused. The manuscript is well-drafted however, there are a few points that must be addressed. Please see below: Line 156-160: The writing pattern changes a lot in Method section, from “After coding the final focus group, codes were discussed within the wider research team (VW, HW, and SY).” To “We decided to group codes into arguments in favour and against claims-based A&F. Then, authors VW and HW re-analysed each transcript and labelled all codes as arguments in favour or against claims-based A&F and identified supporting quotes.” Sticking to one pattern is suggested- either make it interactive (using ‘We’) or make it third narration- (‘the authors’.) Line 173: How do you justify 2 participants for a “Focused Group Discussion’ design? How informative it was to include the FGD with only 2 participants? Line 317: There is still a comment left unaddressed from one of the authors, in the manuscript. What are the unique findings of the study? How can this be applied to the included settings?
---

REVIEWER	Hwang, A. Massachusetts General Hospital
REVIEW RETURNED	04-Dec-2023

GENERAL COMMENTS	This is a qualitative study on how medical specialists in Netherlands view claims-based audit and feedback interventions. This paper adds helpful information to the growing literature on factors that are important to making an audit and feedback intervention successful. I would recommend revisions outlined below.  1. It's not accurate to say that the acceptance of claims-based feedback by professions is unknown as noted in the Objectives section of the Abstract. I would suggest removing this part as there are studies that have examined the benefits and limitations of claims-based feedback as stated by the authors in lines 408-410. The study cited below is another example: Gerteis, M. , Peikes, D. , Ghosh, A. , Timmins, L. , O'Malley, A. , Barna, M. , Taylor, E. , Day, T. , Swankoski, K. , Payne, P. & Brown, R. (2018). Uses and Limitations of Claims-based Performance Feedback Reports: Lessons From the Comprehensive Primary Care Initiative. Journal for Healthcare Quality, 40 (4), 187-193. doi: 10.1097/JHQ.000000000000099. 2. COREQ checklist was not completed fully. Manuscript should be updated to include all required information (i.e. items number 4, 5, 7, 8, 18, and 20). For example, whether study participants had an existing relationship with study staff is important as it may have had an impact on the focus group discussion. 3. Add more information on the different type of hospital listed in Table 1. This will provide additional context about the study setting. 4. Please share all the claims-based A&F on CER that was presented during each focus group as supplemental material rather than just one example shown in Figure 1. That will provide a better sense of the type of claims-based A&F intervention that is being discussed with the focus group participants. 5. Table 1 is referenced in line 184, but Table 1 does not contain any information about whether participants were generally for or against claims-based A&F. This information should be added to the table. 6. The quote in lines 211-213 states that the accuracy and value of claims-based A&F would depend on several factors (e.g. administrator, other supplemental data sources). Therefore, this statement does not directly support argument that claims-based A&F is more reliable than other types of A&F. This quote should be removed and replaced with other supporting statements if available. 7. The quote in lines 222-223 does not directly support the argument that claims-based A&F prevents administrative burden. The participant states that claims-based A&F would be valuable if it can be collected easily. Other supporting quotes should replace
--

	this if available. If not, the conclusion made in line 219 would be invalid. 8. I do not think the quote “we are not more catholic than the pope” was interpreted correctly in lines 248. The participant more likely is saying that medical specialists are not going to strictly abide by the rules to achieve the goal set by A&F. Rather, they are going to find a loophole to meet the goal more easily without changing clinical practice. 9. In line 174 under Participants section, add the specific percentage of participants who were familiar with the use of A&F. 10. Add reference/citation for sentence in 394-395 that says “In line with previous research, this suggests being unknown with A&F, makes it unloved.”
--	---

VERSION 1 – AUTHOR RESPONSE

Reviewer: 1

Dr. Ricardo Espinoza González, Clínica de la Universidad de los Andes

Comments to the Author:

This study represent a contribution in the quality improvement strategy. They ask specialist in different fields of surgery, where the claims-based analysis is very important.

The possible limitation for use only Dutch data I believe is secondary because the reality is similar in other countries.

Thank you for your kind words. We agree with you that the claims data is in fact similar in other countries and have described this in the strength and weakness section (line 532-534).

Reviewer: 2

Dr. Megha Sharma, Karolinska Institutet, RD Gardi Medical College

Comments to the Author:

Do medical specialists accept claims-based Audit & Feedback for quality improvement? a focus group study

Thank you for presenting this manuscript focusing on the issue of acceptance of claim-based audit and feedback among medical specialists. It is an important topic, which is usually not focused. The manuscript is well-drafted however, there are a few points that must be addressed. Please see below:

Thank you for your support and for your helpful comments.

- Line 156-160: The writing pattern changes a lot in Method section, from “After coding the final focus group, codes were discussed within the wider research team (VW, HW, and SY).” To “We decided to group codes into arguments in favour and against claims-based A&F. Then, authors VW and HW re-analysed each transcript and labelled all codes as arguments in favour or against claims-based A&F and identified supporting quotes.” Sticking to one pattern is suggested- either make it interactive (using ‘We’) or make it third narration- (‘the authors’.)

Thank you for this observation. We have rewritten this section to use the third narration pattern consistently.

The following sentences were rewritten:

“The wider research team decided to group codes into arguments in favour and against claims-based A&F.” (line 182-183)

“This was done by analysing key quotes and singling out the figures of speech and key words participants used in order to make a case for or against claims-based A&F (e.g. ‘it works, because it presents a mirror’).” (line 188-191)

“*On the basis of this first-order analysis, the underlying discursive strategies which participants adopted (e.g. presenting claims-based A&F as an effective tool for quality improvement) were analysed.*” (line 191-193)

- Line 173: How do you justify 2 participants for a “Focused Group Discussion’ design? How informative it was to include the FGD with only 2 participants?

We agree with the reviewer that two participants are normally insufficient for a focus group discussion. For this particular focus group five participants agreed to participate on the scheduled date, but due to unforeseen circumstances in clinical practice, three participants did not show without prior notice. As two participants were present at the scheduled time, we decided to conduct the focus group as planned. We have included the results of this focus group, as they were congruent with the themes mentioned in the other four focus groups, and therefore we believe it was appropriate to include these results.

We do agree with the reviewer that this could be a limitation, so this has been described in the strength & weakness section of the discussion: “*Lastly, in some focus groups we had a limited number of participants which may have decreased the level of discussion between participants. However, we did reach data saturation over all focus groups, thus we do not believe that additional arguments would have been identified with more participants per focus group.*” (lines 536-540)

- Line 317: There is still a comment left unaddressed from one of the authors, in the manuscript. Our apologies, we have removed this comment from the manuscript.
- What are the unique findings of the study? How can this be applied to the included settings? We have made the unique findings of our study more explicit in the manuscript.
 - (1) First, to the best of our knowledge, this is the first study on claims-based A&F which focusses on physicians working in hospital settings, earlier studies have focused on primary care physicians.
 - To the “strength and limitations of this study” section after the abstract we have added the bullet point: “*Focus on medical specialists’ perspective on claims-based A&F*” (line 67)
 - To the “Strength and weaknesses” section in the discussion we have added the sentence: “*Second, to the best of our knowledge, this is the first study on claims-based A&F focussing on the perceptions of medical specialists.*” (lines 518-519)

- (2) Second, the discourse analysis has revealed that medical specialists’ hold mixed views on the same issues and that they see both advantages and disadvantages of claims-based A&F. E.g. while previous qualitative research has suggested that participants find “reliability” of data important, this is the first study to reveal that participants have different perceptions of the reliability of claims-based A&F: while some participants perceive claims-based A&F as more reliable, others perceive it as less reliable than A&F based on other data. This shows that it is not simply the reliability of the data that is important in determining the acceptability of claims-based A&F, but rather physicians’ perceptions of its reliability. This provides a deeper understanding of what impacts physician acceptance of claims-based A&F, and gives a more nuanced perspective on the advantages and disadvantages of claims-based A&F.

We have made this more explicit by:

- Rephrasing the strengths in the “strength and weakness” section of the discussion to: “*The first strength of this study is the discourse analysis through which we were able to elicit medical specialists’ mixed views on claims-based A&F, which creates a better understanding of medical specialists’ acceptance of claims-based A&F and shows that it is not simply the reliability of the data that determines the acceptability of claims-based A&F, but rather physicians’ perceptions of its reliability.*” (lines 514-518)
- Rephrasing a sentence in the conclusion to: “*Our study showed that perceptions of medical specialists on A&F in general and claims-based A&F for QI varied, and acceptance of claims-based A&F can be shaped by how A&F is implemented into clinical practice.*” (lines 544-574).

The knowledge of our study can be used to design and implement claims-based A&F in hospitals, which was described in the conclusion:

“*Our study showed that (...) acceptance of claims-based A&F can be shaped by how A&F is implemented into clinical practice.*” (lines 544-574)

“*Thus, policymakers, managers and professionals should consider whether the clinical topic can be represented by claims-based A&F or whether it is necessary to collect other data for A&F with the aim of improving quality.*” (lines 548-550).

Reviewer: 3

Dr. A. Hwang, Massachusetts General Hospital

Comments to the Author:

This is a qualitative study on how medical specialists in Netherlands view claims-based audit and feedback interventions. This paper adds helpful information to the growing literature on factors that are important to making an audit and feedback intervention successful. I would recommend revisions outlined below.

Thank you for your suggestions and for seeing value in the paper.

1. It's not accurate to say that the acceptance of claims-based feedback by professions is unknown as noted in the Objectives section of the Abstract. I would suggest removing this part as there are studies that have examined the benefits and limitations of claims-based feedback as stated by the authors in lines 408-410. The study cited below is another example:

We agree that there are some studies available in which acceptance of claims-based A&F by professionals is discussed. So, we removed this statement. We have rephrased the objective in the abstract as follows:

While claims-based feedback has been previously used for A&F interventions, its acceptance by medical specialists is largely unknown. (lines 33-34)

Second, we have rephrased the objective in the background as follows:

"Research on whether hospital specialists accept the use of claims-based A&F in the context of quality improvement is limited. Aim of this study was to examine whether medical professionals accept claims-based A&F intended for QI purposes." (lines 93-94)

Third, we have added the reference the reviewer suggested to the references cited in the discussion (line 467).

Lastly, we have updated the title of the manuscript to: *"Do medical specialists accept claims-based Audit & Feedback for quality improvement? a focus group study"* (lines 1-2)

2. COREQ checklist was not completed fully. Manuscript should be updated to include all required information (i.e. items number 4, 5, 7, 8, 18, and 20). For example, whether study participants had an existing relationship with study staff is important as it may have had an impact on the focus group discussion.

We have completed the COREQ checklist and added the following information to the manuscript, in the method section.

Under the paragraph "participants":

- *"Reasons for doing the research were explained in the recruitment email and at the start of each focus group."* (lines 127-128)

Under the paragraph "data collection":

- *"Researcher HW (MD-PhD, female) or SY (PhD in organization sciences, male) moderated the sessions. Researcher VW (MD-PhD candidate, female) acted as an observer, presented the examples of claims-based A&F for CER trials and asked clarifying questions when necessary."* (lines 143-146)
- *"Both moderators had extensive experience with focus group research and moderation of focus groups. The observer had prior experience with qualitative research, but not with moderation of focus groups."* (lines 146-152)
- *"Three participants of one focus group (DART) had a previous professional connection to one of the moderators, who had a limited supporting moderation role in this focus group."*

None of the other participants had previous connections to the researchers.” (lines 152-155)

- *Field notes were taken by the observer during each focus group. (line 166)*
- *No repeat focus groups were carried out. (line 167)*

3. Add more information on the different type of hospital listed in Table 1. This will provide additional context about the study setting.

We have included the following explanatory information of the different types of hospitals in the Netherlands to the legenda of Table 1:

*“*In the Netherlands, general hospitals are non-teaching hospitals. Top-clinical hospitals distinguish themselves from general hospitals by meeting certain criteria, which include teaching programs, an infrastructure for innovation and scientific research, and excelling in areas of clinical care. Academic hospitals are large teaching and research hospitals, which provide tertiary care.(18)” (lines 212-216)*

4. Please share all the claims-based A&F on CER that was presented during each focus group as supplemental material rather than just one example shown in Figure 1. That will provide a better sense of the type of claims-based A&F intervention that is being discussed with the focus group participants.

Thank you for this suggestion. We have added the claims-based A&F on CER which was presented during the focus groups to the Supplementary Files, Supplement 4.

5. Table 1 is referenced in line 184, but Table 1 does not contain any information about whether participants were generally for or against claims-based A&F. This information should be added to the table.

We added this information in a separate table to the supplementary files (supplementary file 5, as below). We were unfortunately unable to add this information to table one, as the number of columns would be too high to publish in the main document.

Table 2: Participants' attitudes towards A&F

Focus group	Participant	Attitude towards A&F*	Attitude towards claims-based A&F	Attitude towards A&F developed for this CER study
DART	D1	+/-	+/-	-
	D2	+/-	+/-	NS
	D3	+	-	-
	D4	+	+/-	NS
	D5	+	+	NS
	D6	+/-	+/-	+/-

	D7	+/-	+/-	-
STONE	S1	+	+	+
	S2	-	-	+/-
	S3	+	+	+
	S4	NS	NS	-
Proclion	P1	+/-	-	-
	P2	+/-	-	-
MIRA2	M1	+	+/-	-
	M2	+	NS	-
	M3	NS	-	-
	M4	+	NS	-
	M5	+	-	-
CAPP	C1	+	+/-	+/-
	C2	+	+/-	+
	C3	+	+	+

* (+)=positive attitude, (+/-)=mixed attitude, (-)=negative attitude, NS=not specified

6. The quote in lines 211-213 states that the accuracy and value of claims-based A&F would depend on several factors (e.g. administrator, other supplemental data sources). Therefore, this statement does not directly support argument that claims-based A&F is more reliable than other types of A&F. This quote should be removed and replaced with other supporting statements if available.

Thank you for this observation. We believe that we did not explain this quote correctly in the original manuscript. In the quote, the participant says “now”, referring to the fact that A&F in his current experience is based on data from quality registrations, which are self-registered outcome and complications data by the physicians. Furthermore, with the “data of other sources” the participant was referring to claims data. For correct interpretation, we have therefore reformulated the quote as follows: “*In quality-registry data, it depends on the administrator whether A&F is accurate, if you can improve that by using claims-based A&F, then you have a better and more accurate registration, which gives better A&F.*” (lines 250-252).

This quote then does support the argument that claims-based A&F is more reliable than other types of A&F.

7. The quote in lines 222-223 does not directly support the argument that claims-based A&F prevents administrative burden. The participant states that claims-based A&F would be valuable if it can be collected easily. Other supporting quotes should replace this if available. If not, the conclusion made in line 219 would be invalid.

We understand your point. While the quote is clearly in support of claims-based A&F, the administrative burden is indeed a precondition for successful A&F, rather than an effect of it. We have searched our dataset for alternative quotations and found no other clear quotes

supporting this statement, thus we have decided to omit this point. We have removed this point from the abstract (line 44), results (line 257) and discussion (line 438) section.

8. I do not think the quote “we are not more catholic than the pope” was interpreted correctly in lines 248. The participant more likely is saying that medical specialists are not going to strictly abide by the rules to achieve the goal set by A&F. Rather, they are going to find a loophole to meet the goal more easily without changing clinical practice.

We agree with the reviewer the phrasing of this interpretation was unfit. Thus, we have followed the reviewers suggestion and rephrased the interpretation of this quote as follows: “*Here, the participant used the metaphor “we are not more catholic than the pope” to describe that medical specialists are not going to strictly abide by the rules to achieve the goal set by A&F, but rather find a loophole to meet the goal without having to change their clinical behavior.*” (lines 296-299).

9. In line 174 under Participants section, add the specific percentage of participants who were familiar with the use of A&F.

We have added the specific percentages, by including the following sentences: “*Most participants described familiarity with A&F, with 57% describing regular use, and an additional 24% indicating some level of exposure. Conversely, 19% reported no prior experience with A&F. Specifically focusing on claims-based A&F, 38% of participants described regular use, 14% described some level of exposure and 48% described no prior experience with this type of A&F.*” (lines 204-208)

10. Add reference/citation for sentence in 394-395 that says “In line with previous research, this suggests being unknown with A&F, makes it unloved.”

We have added the references supporting this statement, these are:

- Rouleau G, Reis C, Ivers NM, Desveaux L. Lipstick on a pig: Understanding efforts to redesign audit and feedback reports for primary care . 2022;1–16.
- Desveaux L, Ivers NM, Devotta K, Ramji N, Weyman K, Kiran T. Unpacking the intention to action gap: a qualitative study understanding how physicians engage with audit and feedback. Implementation Science. 2021;16(1):1–9. (lines 449-450)

VERSION 2 – REVIEW

REVIEWER	Hwang, A. Massachusetts General Hospital
REVIEW RETURNED	12-Jan-2024
GENERAL COMMENTS	The authors have done a great job addressing all the comments. I suggest accepting the paper for publication.